# Differential Effects of Patient Navigation across Latent Profiles of Barriers to Care among People Living with HIV and Comorbid Conditions

**DOI:** 10.3390/jcm12010114

**Published:** 2022-12-23

**Authors:** Sharleen M. Traynor, Renae D. Schmidt, Lauren K. Gooden, Tim Matheson, Louise Haynes, Allan Rodriguez, Michael Mugavero, Petra Jacobs, Raul Mandler, Carlos Del Rio, Adam W. Carrico, Viviana E. Horigian, Lisa R. Metsch, Daniel J. Feaster

**Affiliations:** 1Clinical Trials Research Associate Program, Durham Technical Community College, Durham, NC 27703, USA; 2Department of Public Health Sciences, University of Miami Miller School of Medicine, 1120 Northwest 14th Street, Miami, FL 33136, USA; 3Sociomedical Sciences Mailman School of Public Health, Columbia University, 722 West 168th Street, New York, NY 10032, USA; 4Center on Substance Use and Health, San Francisco Department of Public Health, San Francisco, CA 94102, USA; 5Department of Psychiatry and Behavioral Science, Medical University of South Carolina, Charleston, SC 29425, USA; 6Division of Infectious Diseases, University of Miami Miller School of Medicine, Miami, FL 33136, USA; 7Department of Medicine, Division of Infectious Diseases, University of Alabama at Birmingham, 1900 University Blvd # 229, Birmingham, AL 35233, USA; 8Center for Clinical Trials Network, National Institute on Drug Abuse, Rockville, MD 20892, USA; 9Division of Therapeutics and Medical Consequences, National Institute on Drug Abuse, National Institutes of Health, Bethesda, MD 20892, USA; 10Division of Infectious Diseases, Emory University, Atlanta, GA 30322, USA

**Keywords:** HIV, substance use, patient navigation, co-occurring disorders, barriers to care, social determinants of health, syndemic framework

## Abstract

Engaging people living with HIV who report substance use (PLWH-SU) in care is essential to HIV medical management and prevention of new HIV infections. Factors associated with poor engagement in HIV care include a combination of syndemic psychosocial factors, mental and physical comorbidities, and structural barriers to healthcare utilization. Patient navigation (PN) is designed to reduce barriers to care, but its effectiveness among PLWH-SU remains unclear. We analyzed data from NIDA Clinical Trials Network’s CTN-0049, a three-arm randomized controlled trial testing the effect of a 6-month PN with and without contingency management (CM), on engagement in HIV care and viral suppression among PLWH-SU (*n* = 801). Latent profile analysis was used to identify subgroups of individuals’ experiences to 23 barriers to care. The effects of PN on engagement in care and viral suppression were compared across latent profiles. Three latent profiles of barriers to care were identified. The results revealed that PN interventions are likely to be most effective for PLWH-SU with fewer, less severe healthcare barriers. Special attention should be given to individuals with a history of abuse, intimate partner violence, and discrimination, as they may be less likely to benefit from PN alone and require additional interventions.

## 1. Introduction

Despite major progress in the effectiveness and availability of antiretroviral therapy (ART), considerable challenges in the treatment of people living with HIV (PLWH) remain. There are significant gaps in the HIV care continuum, with the greatest deficits seen in retaining individuals in care and achieving viral suppression. Of the 1.1 million individuals living with HIV in the U.S., the Centers for Disease Control and Prevention (CDC) estimates that only 49% are continuously engaged or retained in HIV care and 53% have reached viral suppression [1]. One population that is particularly difficult to engage in care comprises people living with HIV who report substance use (PLWH-SU). In a 2018 review examining predictors of outcomes along the HIV care continuum, substance use was the most commonly cited risk factor associated with poor retention [2]. Compared to PLWH who report no substance use, those who use substances are less likely to access antiretroviral therapy, less likely to adhere to medication plans, and more likely to fall out of care [3]. Psychiatric comorbidities were also found to be predictors of poor retention in HIV care [4,5], and co-occurring diagnoses or “dual disorders”—the presence of both a substance use disorder and at least one psychiatric disorder [6]—among PLWH further complicate clinical management and deter retention in treatment [7]. This can lead to uncontrolled infection, which contributes to ongoing disease transmission. The need for interventions to improve engagement of this high-risk population in HIV care remains a national public health priority [8].

To understand why PLWH-SU do not engage in care, it is necessary to recognize the challenges they face in accessing care. This requires a thorough examination of multi-level factors, including social determinants of health, associated with access to and retention in HIV care. Data from the 2016 sample of the Medical Monitoring Project, a nationally representative sample of all adults diagnosed with HIV in the U.S., showed that 42% of respondents had household incomes below the federal poverty threshold and 43% were unemployed [9]. These factors have consistently been shown to severely limit the resources available to obtain healthcare [10,11,12]. Additionally, 22% and 26% of respondents reported symptoms of depression and anxiety, respectively, both of which are associated with lowered healthcare utilization and poor adherence to treatment plans [9,13,14,15]. Other barriers to HIV treatment identified through both qualitative and quantitative studies include housing instability [11,16,17,18,19], food insecurity [16,20,21], transportation [18,22], substance use [2,23,24], intimate partner violence (IPV) [25,26], perceived stigma [23], discrimination [27], clinic location and hours [24], service availability [26], and privacy concerns [24,28]. Self-perceived barriers to care, including financial, structural, and logistical barriers, as well as concerns about personal health or service delivery, have been associated with higher rates of mortality among PLWH [29]. For PLWH-SU, their substance use presents additional barriers to care, such as stigma, incarceration, and difficulty maintaining scheduled treatment regimens [30,31,32,33].

One intervention specifically designed to help individuals overcome barriers to care is patient navigation (PN). PN is a patient-centered intervention that identifies strategies to eliminate barriers to care and guides individuals through the healthcare system. It employs a strengths-based case management approach [34] and motivational interviewing [35] to empower individuals to manage their healthcare. Examples of activities involved in PN include helping individuals obtain health insurance, scheduling medical appointments, arranging transportation or childcare services, and providing assistance in applying for social services [36]. Screening for social determinants of health, such as housing, is another critical navigation activity [37]. PN may be combined with other tools such as contingency management (CM), which offers financial incentives for completing various activities of a treatment plan. The use of incentives has been specifically useful for engaging people who use drugs and/or alcohol [38,39]. PN strategies can help individuals with layered and complex mental health and/or addictions overcome barriers to obtaining services from various, and often fragmented, systems [40,41].

Although PN was initially developed to help predominantly underrepresented minority women access breast cancer screening and treatment services, it is now used for a variety of patient populations [42,43]. Among PLWH, results of studies assessing the effect of PN on HIV care have been mixed. Some studies have shown that PN is efficacious for linking individuals to care and improving steps along the HIV care continuum [34,44,45,46], while others reported no effect [47,48]. These conflicting results have raised questions about the effectiveness of PN and warrant additional research to determine for which populations and in what circumstances PN is most beneficial [49]. The effectiveness of PN has been difficult to establish, in part because PN is classified as a “complex intervention.” This means that it consists of several core components, targets multiple behaviors, and is often tailored to meet specific conditions [50,51]. The many sources of variation make it difficult to determine which aspects of the intervention are most beneficial. In the case of PN, intervention activities are highly dependent on the specific barriers encountered by each individual. Therefore, it is possible that PN works better for certain patients than for others, depending on the type and number of barriers experienced.

It is also possible that the combination of healthcare barriers, stemming from social determinants of health and other individual and interpersonal factors, may influence the effectiveness of an intervention such as PN on HIV care. Because many PLWH face multiple, concurrent barriers, there is a growing number of studies supporting the use of a syndemics framework for describing factors that influence HIV infection. The syndemics approach suggests that there is an overlap of interrelated factors that drives risk for multiple, co-occurring conditions [52]. Previous studies have identified several syndemic conditions known to occur among PLWH; these include mental illness, violence, homelessness, and socioeconomic disadvantage [53,54]. Previous research has shown that PLWH have an average of two to four syndemic conditions, but some may experience as many as eight conditions [55]. Studies have also shown that these concomitant syndemic factors act synergistically to produce poor health outcomes. For example, research on the interplay among psychosocial factors, substance use, and HIV risk-taking suggests that psychological problems and substance use interact to not only negatively impact retention in care [2], but are also associated with increased risky behaviors [56]. The clustering of risk factors creates a syndemic vulnerability that places individuals at increased risk for HIV acquisition, high-risk sexual behaviors, sexually transmitted infections, and more frequent substance use [52,57].

Additional work has been done to examine syndemic vulnerability as it relates to the HIV care continuum. Glynn et al. (2019) found that among PLWH in Miami, Florida, the odds of having low ART adherence (<80%) and unsuppressed viral load increased for every syndemic condition experienced [55]. This finding supported previous work showing that a higher number of syndemic factors is associated with poor medication adherence and lower odds of viral suppression [58,59]. In 2015, Mizuno et al. (2015) examined syndemic factors specifically among persons who inject drugs and found similar results [57]. All outcomes along the HIV care continuum worsened as the number of psychosocial risk factors increased.

Despite the insights of this previous research, it is limited, in that syndemic barriers to care are measured as a sum of the number of syndemic factors an individual experiences. This composite-score approach places equal weight on the influence of each risk factor and suggests that simply minimizing the number of barriers can lead to improved outcomes. It is possible, however, that the pattern of factors an individual faces is more important than simply the number of barriers. Some syndemic factors may be more significant barriers to treatment than others. Some barriers may be more likely to cluster together than other barriers, creating subgroups of individuals characterized by different combinations of healthcare barriers. Thus, examining patterns of experienced barriers and the impact of these patterns on subsequent health outcomes may provide an improved understanding of how individuals respond to interventions designed to address healthcare barriers.

The current study has two main objectives. The first objective is to describe subgroups of PLWH-SU that share common patterns of barriers to care. The second objective is to analyze how subgroup membership influences the association between PN interventions and HIV outcomes. The data for this study come from CTN-0049, a randomized, controlled trial that studied PN in a sample of 801 hospitalized PLWH-SU who had uncontrolled HIV [60]. The trial tested the effect of a 6-month PN intervention, offered with and without CM, on engagement in care and viral suppression at 6 and 12 months. The results showed that the PN and PN+CM interventions were effective for engaging participants in care at 6 months, and PN+CM was effective for viral suppression at 6 months. Although these effects were not maintained through the 12-month follow-up period, CTN-0049 provides a unique opportunity to explore factors that contributed to the short-term success of the intervention. The data from this study may help characterize the populations likely to benefit from PN; this characterization can inform future adjustments to the intervention, maximize its effectiveness, and result in a more efficient allocation of resources. 

## 2. Materials and Methods

### 2.1. CTN-0049 Overview

The CTN-0049 study was a randomized, controlled trial supported by the National Institute on Drug Abuse’s National Drug Abuse Treatment Clinical Trials Network and has been described in detail elsewhere [61]. Briefly, the purpose of CTN-0049 was to determine the effect of a structured PN intervention, delivered with or without CM, on HIV health outcomes among hospitalized PLWH-SU with advanced HIV disease. Participants were recruited between July 2012 and January 2014 from 11 U.S. hospitals with both a high HIV inpatient census and a high prevalence of substance use among patients. Patients were eligible for enrollment if they had a clinical indication that they were out of HIV care and had evidence of substance use in the past 12 months. A total of 801 participants were randomized to one of three treatment groups: (1) PN, (2) PN+CM, or (3) treatment as usual (TAU). Those randomized to one of the PN groups were offered up to 11 PN sessions over a 6-month intervention period. During sessions, navigators used a strengths-based case management approach to assist patients to coordinate care with clinicians, review their health information, address personal challenges, and provide direct psychosocial support. Those in PN+CM also received financial incentives for target behaviors, including session attendance, completion of paperwork, HIV clinic visits, SUD treatment visits, negative substance use specimens, blood draws, and active ART prescriptions. Participants in the TAU group did not interact with the patient navigators, and received the standard treatment provided at their hospital for linking hospitalized patients to outpatient HIV care and substance-use-disorders treatment, which at most hospitals was written referral. Patients were followed up at 6 months (*n* = 761) and 12 months (*n* = 752) post-randomization and assessed for HIV viral load and other outcomes; however, no differences in rates of HIV viral suppression or death among the three groups at 12 months were revealed.

### 2.2. Measures

#### 2.2.1. Main Outcomes

This study examined how different barrier profiles influenced the effect of PN interventions on four separate outcomes—engagement in HIV care at 6 months, engagement in HIV care at 12 months, viral suppression at 6 months, and viral suppression at 12 months. Engagement in care was measured as a binary variable. Participants were considered “in care” if they self-reported affirmative responses to two questions: “During the past 6 months, did you go to any hospital clinic, hospital outpatient department, community clinic or neighborhood health center for medical care, for example, to care for your HIV/AIDS or other physical problems?” and “If Yes, were any of these HIV primary care visits?” HIV viral load was clinically measured from blood drawn at the 6 and 12-month study visits, or as abstracted from medical records if patients did not attend these visits. The outcome was treated as a binary variable, with a viral load ≤ 200 copies/mL defined as “suppressed” and a viral load > 200 copies/mL defined as “unsuppressed”.

#### 2.2.2. Demographics

Demographic variables were collected at baseline and used in the analysis as follows: age (in years; continuous), race (Black/White; binary), ethnicity (Hispanic/non-Hispanic; binary), and gender (male/female; binary). Education was measured as a categorical variable with these options: middle school or less, some high school/no diploma, high school diploma/GED, junior college, technical/trade/vocational school, some college, college graduate, or graduate/professional school. Categories of race, ethnicity, gender, and education were established in the primary outcomes paper for CTN-0049 [60]. Southern/non-southern residence was a binary variable determined by the study site location [61]. Sites in Atlanta, Baltimore, Birmingham, Dallas, and Miami were considered southern sites. Sites in Boston, Chicago, Los Angeles, New York, Philadelphia, and Pittsburg were considered non-southern sites.

#### 2.2.3. Psychiatric History

Participants were classified as having a psychiatric history if either of two criteria were met: (1) an initial hospital intake (at time of enrollment) with a primary diagnosis and/or any comorbid diagnoses that included terms or conditions such as “suicidal ideation”, “psychosis”, “schizophrenia”, “bipolar disorder”, “PTSD”, “hallucinations”, “mood disorder”, and “altered mental state,” or (2) participant self-report that they “saw a professional for the primary purpose of getting help for psychological or emotional issues in the past 6 months”.

#### 2.2.4. Barriers to Care

An analysis of barriers to care was guided by a socioecological framework described by Mugavero et al. (2013) to examine engagement in HIV care across multiple levels of healthcare access [62]. Building upon earlier models of healthcare utilization, this framework categorizes healthcare utilization factors into four categories: (1) Individual factors, which may include demographics, personal health beliefs, past experiences, and coping skills; (2) Relationship factors, which may include connections with family, friends, and medical providers; (3) Community/health system factors, which may include community-level poverty, social norms, and the local health service infrastructure; and (4) Policy factors, which may include treatment guidelines, service coordination, and funding. This study specifically examined 23 barriers to care at the first two levels (individual and relationship factors) and health system factors at the third level. Addressing community factors and policy-level barriers was beyond the scope of this research. All measures were assessed at baseline.

I.Alcohol use severity—This was measured on a continuous scale using the Alcohol Use Disorders Identification Test (AUDIT) [63]. This is a 10-item questionnaire assessing the frequency of alcohol consumption, alcohol dependence, and harmful consequences of alcohol use. Each item was scored on a scale from 0 to 4, with a total score range from 0 to 40. A sample question is, “How often during the last year have you failed to do what was normally expected of you because of drinking?” (0 = never, 1 = less than monthly, 2 = monthly, 3 = weekly, 4 = daily or almost daily). Higher scores represent greater alcohol use severity.II.Drug use severity—This was measured on a continuous scale using a short version of the Drug Abuse Screening Test, the DAST-10 [64,65]. This is a 10-item questionnaire with “yes” or “no” response options for each item. A sample item is, “Have you had ‘blackouts’ or ‘flashbacks’ as a result of drug use?” All items with a “yes” response represent 1 point on a total scale from 0 to 10. Greater scores represent greater drug use severity.III.Food insecurity—This was measured on a continuous scale using the Household Food Security Access Scale [66]. This is a 9-item questionnaire assessing various food insecurity domains, such as quantity, quality, and uncertainty experienced in the past 4 weeks. Each item was scored from 0 to 3 based on the frequency of experiencing each domain. For example, “In the past four weeks, did you worry that your household would not have enough food? How often did this happen?” (0 = never, 1 = rarely, 2 = sometimes, 3 = often). Total scores ranged from 0 to 27, with higher scores representing greater food insecurity.IV.History of abuse—This was measured as a binary variable. Participants who reported any history (either as a child or an adult) of being beaten, physically attacked or abused, raped, or sexually abused were scored a 1. Others were scored a 0.V.History of IPV—This was measured as a binary variable and was based on 4 “yes/no” items from a previously published IPV screening tool [67]. A sample question is, “Have you ever been in a relationship where a sexual partner threw, broke, or punched things?” Participants who answered affirmatively to any of the items were scored a 1. Others were scored a 0.VI.Recent incarceration—This was measured as a binary variable and was based on participant self-report of being incarcerated in the past 6 months.VII.Housing insecurity—This was measured as a binary variable. Participants who self-reported being homeless or living in a shelter, transitional housing, hotel, group home, or other residential facility in the last 6 months were scored a 1. Others were scored a 0.VIII.Language barriers—This was measured as a binary variable and was based on participant self-report as to whether English was their second language.IX.Lack of health insurance—This was measured as a binary variable and was based on participant self-report of current health insurance status.X.Lack of a case manager—This was measured as a binary variable and was based on participant response to the question, “During the past 6 months, did you receive any help from case managers or social service workers with things like obtaining health care or legal services, housing, or easing money problems?”XI.Lack of transportation—This was measured as a binary variable based on participant self-report about how they got to their most recent medical appointment. If participants indicated that they drove themselves, they were scored a 0. Others who, for example reported taking public transportation, being taken by somebody else, or walking, were scored a 1.XII.Low access to healthcare—This was measured as a continuous variable using a 6-item instrument that was adapted from an instrument assessing medical care for low-income persons with HIV [10]. Each response was scored on a scale from 0 to 4, for a total score range from 0 to 24. Higher scores represented lower access to care, and in some cases, items were reverse-scored to maintain this pattern. A sample item is, “I am able to get medical care whenever I need it” (0 = strongly agree, 1 = somewhat agree, 3 = uncertain, 4 = somewhat disagree, 5 = strongly disagree).XIII.Low health literacy—This was measured as a continuous variable using a brief 3-item health literacy screening tool [68]. Each response was scored on a scale from 0 to 4, for a total score range from 0 to 12. Items were reverse-scored so that higher scores represented lower health literacy. A sample question is, “How confident are you filling out medical forms by yourself?” (0 = extremely, 1 = quite a bit, 2 = somewhat, 3 = a little bit, 4 = not at all).XIV.Low income—This was measured as a binary variable based on participant self-report of income level according to categories of income range. Participants with incomes less than $10,000/year were considered low-income. This cut point was chosen based on poverty thresholds determined by the 2014 U.S. Census Bureau, which was $12,071 for a single person [69].XV.Low readiness for substance use treatment—This was measured as a continuous variable using 4 items derived from a previously published treatment readiness instrument. [70]. Each item was scored on a scale from 1 to 5, for a total score range from 4 to 20 Items were reverse-scored so that higher scores represented lower readiness for treatment. A sample item is, “You want to be in a treatment program” (1 = strongly agree, 2 = agree, 3 = undecided, 4 = disagree, 5 = strongly disagree).XVI.Low perceived health status—This was measured as a continuous variable using the SF-12 instrument, a 12-item short form health survey [71]. Ten items were scored on a scale from 1 to 5, and two items were scored on a scale from 1 to 3, for a total score range from 12 to 56. Items were scored so that higher scores represented lower perceived health. A sample item is, “Does your health now limit you in moderate activities such as moving a table, pushing a vacuum cleaner, bowling, or playing golf?” (1 = no, not at all, 2 = yes, limited a little, 3 = yes, limited a lot).XVII.Low social support—This was measured as a continuous variable based on responses to 5 items adapted from a social support instrument for HIV-infected individuals measuring support over the last 4 weeks [72]. Each item was scored on a scale from 1 to 5, for a total score range from 5 to 25. Lower scores represented lower social support. A sample item is, “How often was someone to love and make you feel wanted available to you during the past 4 weeks if you needed it?” (1 = none of the time, 2 = a little of the time, 3 = some of the time, 4 = most of the time, 5 = all of the time).XVIII.Medical mistrust—This was measured as a continuous variable using the Group-Based Medical Mistrust Scale [73]. Each of the 12-items were scored on a scale from 1 to 5, for a total score range from 12 to 60. Higher scores represented greater medical mistrust, and some items were reverse-scored to maintain this pattern. A sample item is, “Doctors and health care workers sometimes hide information from patients who belong to my ethnic group” (1 = strongly disagree, 2 = disagree, 3 = neither agree nor disagree, 4 = agree, 5 = strongly agree).XIX.History of discrimination—This was measured as a binary variable. Participants who self-reported that they had ever experienced discrimination, been prevented from doing something, been hassled, or made to feel inferior in a healthcare setting because of their gender, sexual orientation, race, ethnicity, HIV status, or drug use were scored a 1. Others were scored a 0.XX.Social conflict—This was measured as a continuous variable based on responses to 3 items adapted from a Conflictual Social Interactions instrument measuring conflict over the last 4 weeks [72]. Each item was scored on a scale from 1 to 5, for a total score range from 5 to 15. Higher scores represented greater social conflict. A sample item is, “During the past 4 weeks, how much of the time have you had serious disagreements with your family about things that were important to you?” (1 = none of the time, 2 = a little of the time, 3 = some of the time, 4 = most of the time, 5 = all of the time).XXI.Psychological distress—This was measured as a continuous variable using the 18-item Brief Symptom Inventory to assess depression, anxiety, and somatization [74]. Each item was scored on a scale of 0 to 4 with higher scores representing greater psychological distress. The three domains were combined into a single score for a total score range of 0 to 72. A sample item is, “In the past 7 days, how much were you distressed by feeling lonely?” (0 = not at all, 1 = a little bit, 2 = moderately, 3 = quite a bit, 4 = extremely).XXII.Negative attitudes toward substance use treatment—This was measured as a continuous variable using a 4-item subscale of the Treatment Attitude Profile [75]. Each item was scored on a scale of 1 to 5, for a total score range from 4 to 20. Higher scores represented greater negative attitudes toward treatment. A sample item is, “Substance use treatment programs have too many rules and regulations for me” (1 = strongly disagree, 2 = disagree, 3 = undecided, 4 = agree, 5 = strongly agree).XXIII.Unemployment—This was measured as a binary variable based on participant self-report that they were unemployed.

### 2.3. Statistical Analyses

First, a latent profile analysis (LPA) was conducted to identify subgroups of individuals with similar barriers to care. LPA is a latent variable modeling technique that identifies unobserved subgroups of individuals within a population based on responses to a set of observed variables; it assumes that individuals can be categorized by patterns of responses that relate to profiles of personal and/or environmental attributes [76]. LPA, rather than a Latent Class Analysis, was used in this analysis, as it can accommodate both categorical and continuous indicators [77].

The current study included the 23 barriers to care previously described. Profile solutions were evaluated based on several standard fit indices, including Akaike information criteria (AIC), adjusted Bayesian criteria (BIC), model entropy, Lo–Mendel–Rubin test, and the bootstrapped likelihood-ratio test. Additionally, the clinical meaningfulness, interpretability, and sample size of each class were considered in the selection of the final model. Latent profile plots were created to visualize differences between the profiles. Differences in latent profiles by gender, race, and southern/non-southern residence were assessed using a likelihood-ratio test with a significance level of α = 0.05. 

Next, structural models were constructed to test how the relationship between the intervention groups (PN, PN+CM, and TAU) and the four distal outcomes of interest differed by profile. Model construction followed a 3-step approach [66,67]. In step 1, LPA was performed; age, gender, southern/non-southern residence, and treatment group were included as covariates using the auxiliary option in Mplus. In step 2, a new latent profile variable was created by incorporating the classification error obtained from the step 1 logits for classification probabilities. This classification method is preferred over other methods such as classify–analyze or pseudo-class draw approaches because it accounts for uncertainty in latent profile assignment and reduces bias [78]. In step 3, the distal outcome was regressed on the intervention variables, controlling for the covariates and comparing effects across the latent profiles. This process was repeated for each outcome. Odds ratios were used to interpret the effect of latent profile on each outcome.

## 3. Results

### 3.1. Characteristics of the Study Population

Select demographic, clinical, psychosocial, and healthcare access factors of the 801 study participants are summarized in Table 1. The sample was mostly male (67.4%), Black (82.5%), and Non-Hispanic (89.0%) with a mean age of 44.2 years. There were slightly more participants enrolled from southern sites (59.2%) than from northern sites. The average time since HIV diagnosis was 11.8 years. Most participants reported a history of being in HIV care (82.9%) and being on antiretroviral therapy (77.2%) at some point in their lives, but approximately two-thirds of the sample had a CD4 count of less than 200 cells/μL at enrollment. About one-third of participants reported injection drug use in the last 12 months, and 55.3% had a history of substance use treatment in the 6 months prior to enrollment. Approximately 22.0% of the study sample had a recorded psychiatric history. The overall baseline mean of psychological stress as measured by the BSI-18 was 22.5 (16.1 SD). Based on established BSI thresholds, there were 39 individuals with minor elevation, 17 with moderate elevation, and 11 with marked elevation [79].

Many participants had achieved at least a high school education (60.2%), but most (77.4%) reported an annual income less than $10,000, and only 11.6% were employed. Most individuals reported unreliable transportation (90.3%), not having a case manager (70.0%), and low levels of social support (mean = 14.7 out of 25). Many participants, however, had health insurance (67.4%) and moderate to high levels of health literacy (mean = 9.0 out of 12). There were no differences in the distribution of baseline characteristics across treatment groups, which was expected due to randomized treatment assignment. The reliabilities of measurement scales are shown in Table 2. All scales had a Cronbach alpha > 0.70, indicating adequate reliability.

### 3.2. LPA Results

Models of two to five profiles were considered for the LPA. The five-profile solution was ruled out because the best likelihood value could not be replicated after 2000 random starts. Among the remaining models, multiple fit statistics (Table 3) and interpretability indicated that a three-profile solution best fit the data. The sample-size adjusted BIC score (69,367.41) was lower in the three-profile solution than the two-profile solution (indicating a better fit), while maintaining a high entropy (0.863). The Lo–Mendell–Rubin adjusted likelihood-ratio test, however, showed that the four-profile solution did not significantly improve fit above the three-profile solution (*p* = 0.166). The three-profile solution also presented a logical substantive interpretation, adequate class distinction, and adequate sample sizes. Therefore, the three-profile solution was selected as the best model.

A comparison of the three profiles is described in Table 4 and displayed in Figure 1. Standardized means are shown for continuous variables, and proportions of item endorsement are shown for dichotomous variables. The first profile had relatively low barriers to care. Values for all barriers were the lowest for this profile except for lack of case management, low income, and not having insurance. This profile comprised half of study participants (50.3%) and was labeled “Lower Barriers (LB).” The second profile, which described 35.7% of the study sample, generally exhibited higher barriers to care compared to the first profile and was characterized by having a higher probability of reporting a history of abuse (67.3%) and intimate partner violence (65.6%). This profile was labeled “Higher Barriers with Abuse and Violence (HB-AV).” The third profile, which comprised 14.0% of the study sample, was quite close to the second, with similar values across most of the barriers. The main distinguishing features of this profile were an even higher likelihood of having a history of abuse (74.8%) and intimate partner violence (65.6%) and a high likelihood of having experienced discrimination (std mean = 5.41). This profile was labeled “Higher Barriers with Discrimination, Abuse and Violence (HB-DAV).” This three-profile solution was further analyzed for differences by key demographic characteristics including gender, race, and southern/non-southern residence (see Appendix A.).

### 3.3. Structural Model Results

Estimates for the final three-profile LPA are shown in Table 5. After controlling for race, gender, and southern/non-southern residence, structural models indicated that there were significant effects of the PN and PN+CM interventions on being engaged in care at 6 and 12 months and viral suppression at 6 months. However, these associations were only observed for certain profiles. The greatest effects were seen for the Lower Barrier (LB) profile, where the PN+CM group was associated with higher likelihood of being in care at 6 months (β = 1.37, OR = 3.94, *p* < 0.001), being virally suppressed at 6 months (β = 0.687, OR = 1.99, *p* = 0.15), and being in care at 12 months (β = 0.881, OR = 2.41, *p* = 0.019), compared to the TAU group. The PN-only group also had a significant effect on viral suppression at 6 months (β = 0.610, OR = 1.85, *p* = 0.035) and a marginally significant effect on being in care at 6 months (β = 0.660, OR = 1.93, *p* = 0.054), compared to the TAU group. The Higher Barriers with Abuse and Violence (HB-AV) profile had higher odds of being engaged in care for both the PN+CM group (β = 1.25, OR = 3.49, *p* = 0.001) and the PN group (β = 0.981, OR = 2.67, *p* = 0.018) compared to the TAU group, but there were no significant associations with the other distal outcomes of interest. The interventions did not have any significant effects for those with the Higher Barriers with Discrimination, Abuse, and Violence (HB-DAV) profile. Additionally, there were no significant intervention effects on viral suppression at 12 months for any of the latent profiles. 

## 4. Discussion

This study provides important insights about the differential effects of PN interventions for engaging PLWH-SU in care. It suggests that PN, offered with or without CM, is most effective for individuals with relatively low levels of healthcare barriers. Of the three barrier profiles identified in this analysis, the LB group had the greatest response to PN, with higher 6- and 12-month rates of engagement in care and viral suppression than the TAU group. The positive intervention effects observed for the LB group may be explained by the absence of extreme healthcare barriers that would delay or compete with the need to engage in care. For example, if a patient has an overwhelming and immediate need to address an aspect of their wellbeing, such as a severe mental health condition or unstable housing, the patient may prioritize such a need over HIV care. The navigator would need to help resolve these other issues before the patient is ready to focus on HIV care. If, however, a patient is stable and does not require other assistance, the patient navigator can focus on linking the individual directly to care.

Conversely, individuals with a history of abuse, IPV, and discrimination are not likely to benefit from stand-alone PN interventions. In this analysis, the HB-AV group had only a partial response to the PN and PN+CM interventions. These individuals had higher odds of being engaged in care at 6 months, but these effects were not sustained at 12 months and did not lead to viral suppression. Finally, the HB-DAV group did not respond to either of the PN interventions. These results imply that PN (with or without CM) is not sufficient for all patient populations and underscore the importance of a thorough assessment of patients’ needs when recommending behavioral interventions. In an era of precision medicine, the development of personalized interventions is becoming increasingly more valuable in prevention science.

These findings contribute to the current science of healthcare utilization among PLWH-SU by identifying high-risk barrier profiles. Specifically, a history of abuse, intimate partner violence, and/or discrimination are important indicators of a high overall level of healthcare barriers. In both profiles characterized by abuse and IPV, nearly all other barriers were present at higher rates compared to the profile without abuse and IPV. This is consistent with other work, most notably Singer’s work on the SAVA syndemic of substance abuse, violence, and HIV/AIDS, indicating that these factors are likely to co-occur [25,26,53,69,80]. Individuals who experience IPV and/or abuse are more likely to suffer from depression and other psychiatric disorders [54] Substance use in this context further perpetuates violence and abuse. Healthcare personnel should be cognizant of these factors, incorporate screening for multiple conditions into practice, and be prepared to link patients to the appropriate programs or provide appropriate co-located services.

This study also provides insights about the potential impacts of trauma and abuse on the effectiveness of health interventions. PN interventions designed to engage PLWH-SU in care were not found to be effective for individuals with a history of IPV, physical or sexual abuse, or discrimination. This finding may be related to the possibility that individuals are still in abusive relationships at the time PN interventions are administered. Individuals in such situations may lack the resources and/or the autonomy to independently seek healthcare or suffer from fear or anxiety about being in a healthcare setting where the abuse may be discovered. Even if an individual is not actively in an abusive situation, the harms from past events may have lingering mental health effects that influence one’s decision to seek care. Additionally, if a person experienced abuse or discrimination in a healthcare setting, this could deter that person from seeking care in the future. Thus, the identification of psychosocial barriers to care is an important part of a routine needs assessment, and it is especially important to determine if there is a history of abuse or IPV, with or without discrimination. Alternatives to PN, or PN delivered in combination with other interventions, may be required to result in positive health outcomes for these individuals.

Another noteworthy finding is that among the subgroups that had positive responses to the PN interventions, the effects were stronger when CM was added to PN, compared to PN alone. It could be that the combination of PN and CM interventions targets both intrinsic and extrinsic motivations for behavior change [70]. Alternatively, the financial incentives may have enhanced the effect of PN by encouraging individuals to attend more PN sessions [81]. Additional research is needed to evaluate the effect of this combined approach on this study’s population and other patient populations.

The results of this study should be considered with the following limitations. First, there may have been some degree of measurement error associated with the barriers to care included in the analysis. All measures were self-reported and some of the barriers were measured indirectly. For example, information about transportation barriers was derived from responses about how participants got to their clinic appointment. A better way would be to specifically ask about transportation barriers to healthcare. Measurement tools designed specifically to evaluate barriers to HIV care, such as the Kalichman’s Barriers to Medical Care instrument [82], should be considered for future studies. Second, this analysis considered only individual and relationship-level barriers to care. A more comprehensive examination that includes higher level barriers, such as system and policy factors, may reveal other distinct profiles that impact the response to interventions and should be explored in future analysis. Finally, the results of this work are limited to a specific population and may not be generalizable to other populations of PLWH. This study’s population was a highly disadvantaged group of individuals with advanced HIV disease. This may have reduced the variability in observed healthcare barriers, as most of the CTN-0049 participants suffered from multiple barriers. Further research is needed to determine if similar barrier profiles exist in other populations.

Despite these limitations, this study has significant implications for public health practice. It underscores the importance of screening PLWH-SU for a history of abuse, IPV, and discrimination. Not only are they indicators of particularly vulnerable individuals, but they may also reduce the effectiveness of otherwise beneficial interventions. If these conditions are present, protocols should be initiated to make the appropriate referrals to mental health or social services. Screenings and follow-up assessments should be an ongoing part of interventions, not just part of the baseline evaluation. While this study was conducted in a U.S. population enrolled in a clinical trial, there may be important considerations for other PLWH populations with co-occurring drug use to identify and meet needs at the complex intersection of substance use and HIV services. A global review of studies assessing the integration of HIV and substance-use services showed that increased service integration can improve patient outcomes among this population across a variety of service models, both in and outside the U.S. [83]. Additionally, strategies to integrate treatment for mental health and substance use disorders among PLWH have been implemented in low-to-middle income countries [84]. Finally, this study builds on existing work by describing the complexities of how healthcare barriers group together. It suggests that, in addition to the number of barriers to care an individual faces, there are specific-barriers profiles that can differentially impact care. As a next step, it would be useful to conduct a direct comparison of latent variable approaches using barrier profiles with the composite-risk score method used in previous studies.

## 5. Conclusions

In summary, this research elucidated the complexities of engaging PLWH-SU in treatment. It identified distinct healthcare barrier profiles among PLWH-SU enrolled in CTN-0049, each with different responses to PN interventions. These results help to inform the use of PN programs and provide a more efficient target for PN resources by more narrowly defining the patient population for which PN is effective.

## Figures and Tables

**Figure 1 jcm-12-00114-f001:**
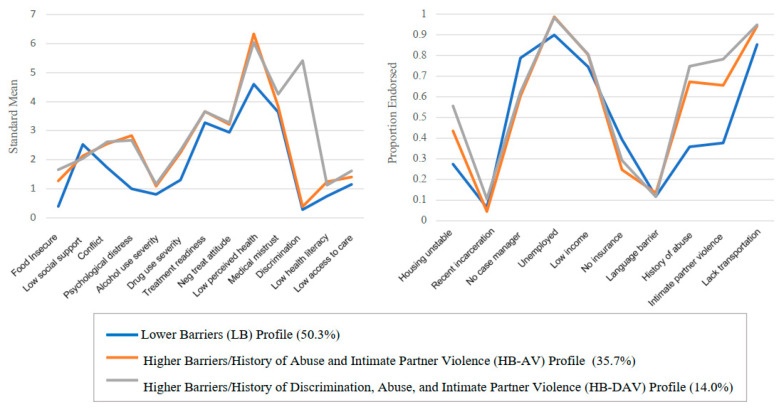
Visualization of Three-Class Latent Profile Analysis solution using 23 indicators of barriers to care.

**Table 1 jcm-12-00114-t001:** Characteristics of the CTN-0049 study sample (*n* = 801).

	Range	Treatment as Usual (*n* = 264)	Patient Navigation (*n* = 266)	Patient Navigation + Contingency Management (*n* = 271)
Demographics				
Age (years)	18–68	44.0 (10.1)	44.3 (9.9)	44.2 (10.0)
Male		184 (69.7%)	179 (67.3%)	177 (65.3%)
Black race		216 (81.8%)	226 (85.0%)	219 (80.8%)
Hispanic ethnicity		35 (13.3%)	28 (10.5%)	25 (9.2%)
Education (high school grad or more)		167 (63.3%)	149 (56.0%)	166 (61.3%)
Southern U.S. residence		155 (58.7%)	158 (59.4%)	161 (59.4%)
Clinical Characteristics				
Baseline CD4 count (cells/μL)	0–1482	152.6 (150.4)	157.5 (168.4)	171.3 (172.3)
Years since HIV diagnosis	0–32	12.1 (8.9)	12.1 (11.0)	11.2 (8.3)
Ever in HIV care		227 (86.3%)	219 (82.6%)	218 (80.4%)
History of antiretroviral therapy		208 (79.1%)	203 (76.3%)	207 (76.7%)
Injection drug use, last 12 months		85 (32.2%)	90 (33.8%)	85 (31.4%)
Substance use treatment, last 6 months		149 (56.6%)	152 (57.1%)	142 (52.4%)
Hepatitis C positive		87 (34.0%)	90 (34.0%)	81 (30.4%)
Psychiatric History		56 (7.0%)	67 (8.4%)	53 (6.6%)
Individual Barriers to Care				
Employed (full-time, part-time, temp)		34 (12.9%)	24 (9.0%)	35 (12.9%)
Low income (<$10,000/year)		166 (77.9%)	181 (80.4%)	171 (74.0%)
Uninsured		85 (32.6%)	88 (33.3%)	88 (32.6%)
Health literacy	0–12	9.2 (3.0)	9.0 (3.2)	8.8 (3.2)
Language barrier		37 (14.0%)	31 (11.7%)	31 (11.4%)
Access to healthcare	0–24	17.8 (4.7)	14.5 (5.0)	18.2 (4.7)
Perceived health status	0–55	33.3 (9.3)	33.8 (9.5)	33.9 (8.7)
Food insecurity	0–27	6.5 (8.2)	6.4 (8.1)	5.8 (7.3)
Housing insecurity		91 (34.5%)	106 (39.9%)	101 (37.3%)
Lack of transportation		199 (87.7%)	211 (92.1%)	229 (90.9%)
Psychosocial distress (BSI-18)	0–69	22.2 (16.1)	23.0 (16.4)	22.4 (15.8)
Alcohol use severity (AUDIT)	0–38	9.2 (9.5)	9.0 (9.7)	8.9 (9.5)
Substance use severity (DAST-10)	0–10	4.6 (2.9)	4.6 (3.0)	4.8 (2.9)
Negative treatment attitudes	4–20	10.7 (3.4)	10.7 (3.5)	10.7 (3.5)
Readiness for treatment	4–20	14.0 (4.4)	14.5 (3.8)	14.1 (4.4)
History of incarceration, last 6 months		16 (6.1%)	20 (7.5%)	15 (6.4%)
Relationship Barriers to Care				
Social support	0–25	14.7 (6.8)	14.6 (6.3)	14.7 (6.4)
Social conflict	0–15	6.9 (3.5)	6.6 (3.3)	6.8 (3.4)
Medical mistrust	12–60	29.1 (7.8)	28.9 (8.0)	28.1 (7.4)
History of discrimination	0–5	0.6 (1.2)	0.6 (1.1)	0.5 (1.0)
History of abuse		134 (50.8%)	129 (48.5%)	158 (58.3%)
History of intimate partner violence		145 (54.9%)	132 (49.6%)	151 (55.72%)
No case manager		188 (71.8%)	188 (70.7%)	182 (67.4%)

Range, mean (std dev) shown for continuous variables; *n* (%) shown for dichotomous variables. Abbreviations: BSI = Brief Symptom Inventory, AUDIT = Alcohol Use Disorders Identification Test, DAST = Drug Abuse Severity Test.

**Table 2 jcm-12-00114-t002:** Reliability of continuous scales used to measure barriers to care.

	Cronbach Alpha
Food insecurity	0.944
Intimate partner violence	0.829
Social support	0.861
Social conflict	0.746
Psychological Distress—(BSI-18)	0.916
Alcohol use severity (AUDIT)	0.864
Drug use severity (DAST-10)	0.824
Readiness for substance use treatment	0.835
Attitudes about substance use treatment	0.747
Perceived health status	0.856
Medical mistrust	0.849
Experienced discrimination	0.718
Health literacy	0.731
Access to care	0.725

Abbreviations: BSI = Brief Symptom Inventory, AUDIT = Alcohol Use Disorders Identification Test, DAST = Drug Abuse Severity Test.

**Table 3 jcm-12-00114-t003:** Latent Profile Enumeration using 23 indicators of barriers to care.

Number of Profiles	Log-Likelihood	AIC	aBIC	Entropy	LMR-A *p*-Value	BLRT *p*-Value
1	−35,383.31	70,838.62	70,892.99	--	--	--
2	−34,802.32	69,724.63	69,815.25	0.802	<0.001	<0.001
3	−34,536.27	69,240.54	69,367.41	0.863	<0.001	<0.001
4	−34,426.40	69,068.80	69,231.91	0.892	0.166	<0.001

Abbreviations: AIC = Akaike information criteria, aBIC = adjusted Bayesian information criteria, LMR-A = Lo–Mendell–Rubin adjusted likelihood-ratio test, BLRT = bootstrapped likelihood-ratio test.

**Table 4 jcm-12-00114-t004:** Standard means and proportions of continuous and categorical indicators by profile.

	Lower Barriers	Higher Barriers, Abuse and IPV	Higher Barriers, Discrimination, Abuse and IPV
Continuous Indicators	Standard Means		
Food insecurity	0.392	1.275	1.655
Social support (higher = more support)	2.528	2.132	2.043
Conflict	1.727	2.547	2.622
Psychological distress	0.995	2.827	2.674
Alcohol use severity	0.811	1.088	1.16
Drug use severity	1.302	2.249	2.33
Readiness for substance use treatment	3.276	3.659	3.659
Negative attitudes about drug treatment	2.945	3.216	3.277
Low perceived health status	4.602	6.334	6.042
Medical mistrust	3.637	3.835	4.263
History of discrimination	0.282	0.387	5.412
Low health literacy	0.751	1.244	1.125
Low access to care	1.154	1.404	1.617
Categorical Indicators	Proportion Endorsed		
Housing instability	27.4%	43.5%	55.6%
Recent incarceration (last 6 m)	6.4%	4.5%	10.6%
No case manager	78.7%	60.3%	62.3%
Unemployment	89.9%	98.7%	98.3%
Low income	74.5%	80.3%	80.6%
Uninsured	39.4%	24.7%	29.3%
Language barrier	11.8%	13.5%	11.7%
History of abuse	35.8%	67.3%	74.8%
History of intimate partner violence	37.7%	65.6%	78.2%
Lack of transportation	85.3%	94.3%	94.8%

**Table 5 jcm-12-00114-t005:** Effect of patient navigation interventions on engagement in care and HIV viral suppression by latent profile.

	Lower Barriers (*n* = 403)			Higher Barriers, Abuse, IPV (*n* = 286)			Higher Barriers, Discrimination, Abuse, IPV (*n* = 112)		
	est.	s.e.	*p*-val	est	s.e.	*p*-val	est	s.e.	*p*-val
Engaged in care—6 months									
PN intervention	0.660	0.342	0.054	0.981	0.413	0.018	0.160	0.706	0.820
PN+CM intervention	1.370	0.387	<0.001	1.250	0.393	0.001	1.881	1.185	0.112
Race	−1.242	0.651	0.056	0.636	0.395	0.107	1.087	0.848	0.200
Age	−0.016	0.014	0.248	−0.010	0.018	0.566	−0.111	0.041	0.007
Gender	0.074	0.325	0.820	−0.209	0.334	0.532	−0.799	0.850	0.347
Southern U.S.	−0.928	0.357	0.009	−0.884	0.357	0.013	−0.426	0.758	0.574
In care at baseline	0.643	0.334	0.054	0.325	0.349	0.352	2.284	0.789	0.004
Viral suppression—6 months									
PN intervention	0.610	0.291	0.035	−0.357	0.392	0.363	−0.318	0.534	0.551
PN+CM intervention	0.687	0.282	0.015	0.337	0.340	0.321	−0.008	0.581	0.988
Race	−0.534	0.331	0.107	−0.704	0.362	0.052	−0.521	0.627	0.406
Age	−0.005	0.011	0.655	0.029	0.017	0.086	0.039	0.024	0.107
Gender	−0.398	0.283	0.160	−0.295	0.305	0.332	0.584	0.498	0.241
Southern U.S.	−0.679	0.236	0.004	0.027	0.297	0.927	−1.107	0.463	0.017
Suppressed at baseline	0.943	0.404	0.019	1.206	0.474	0.011	1.613	0.693	0.02
Engaged in care—12 months									
PN intervention	0.140	0.335	0.676	0.165	0.428	0.700	−0.358	0.584	0.540
PN+CM intervention	0.881	0.376	0.019	0.143	0.392	0.716	0.075	0.789	0.925
Race	−0.495	0.501	0.323	0.095	0.217	0.828	1.576	0.71	0.026
Age	−0.002	0.014	0.891	0.026	0.018	0.142	0.025	0.028	0.377
Gender	−0.225	0.313	0.473	−0.072	0.335	0.829	−0.581	0.627	0.355
Southern U.S.	−0.381	0.322	0.237	−0.577	0.343	0.093	0.634	0.558	0.256
In care at baseline	1.237	0.344	<0.001	0.186	0.34	0.584	0.832	0.542	0.125
Viral suppression—12 months									
PN intervention	0.406	0.282	0.151	−0.249	0.398	0.531	−0.264	0.565	0.640
PN+CM intervention	0.266	0.288	0.356	0.339	0.354	0.338	−0.619	0.614	0.313
Race	−0.927	0.333	0.005	−0.357	0.365	0.328	−1.476	0.722	0.041
Age	0.013	0.012	0.267	0.014	0.017	0.419	0.034	0.031	0.271
Gender	−0.087	0.268	0.746	0.178	0.305	0.559	0.389	0.517	0.451
Southern U.S.	−0.718	0.238	0.003	−0.674	0.309	0.029	−0.787	0.489	0.108
Suppressed at baseline	0.179	0.403	0.656	0.991	0.471	0.035	1.669	0.615	0.007

## Data Availability

CTN 0049 trial data are publicly available on NIDA Data Share: https://datashare.nida.nih.gov/ (accessed on 7 September 2018).

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
