# Peer review of "Differential Effects of Patient Navigation across Latent Profiles of Barriers to Care among People Living with HIV and Comorbid Conditions"

_jcm, 2022, doi:10.3390/jcm12010114_

Round 1
Reviewer 1 Report
Thank you for designing and implementing this very insightful and impactful research. The manuscript is presented nicely and flows easily for comprehensibility. However, I would suggest that authors consider justifying restricting gender classification to the binary groups of male and female. Considering that this study sourced secondary data, it might be useful for readers to know if gender data for all participants is binary and if not, the justification for choosing binary grouping. Another important area to reconsider is the way health literacy was measured. Authors have reported using Cronbach alpha to assess the reliability of their data. It is important to mention that Cronbach alpha is not appropriate for knowledge-based measures, which may have correct and incorrect responses. Overall, this is a very good paper that has the potential of improving the health and wellbeing of people living with HIV.
Reviewer 2 Report
Thank you for this very relevant article elucidating barriers to care among PLWH in a very specific, US-American patient population. While I appreciate the general approach and findings I recommend clarification of statistical methods and clearer presentation of ffindings for a more general audience ("Journal of Clinical Medicine") before publication.
- The statistical methods are hard to follow for a non-statistician. Please consider revising to make this crucial part of the paper more accessible.
One example is Table 3 (and also table 4) which seems entirely in-comprehensible without in-depth understanding of the statistical methods.
- The components of the different intervention arms from the parent study should be reported.
- Although you briefly discuss that results are hard to extrapolate to other patient populations, as a non-American reader it would be very interesting to hear your thoughts on how your findings could translate (policy or further research) to other PLWH populations with high injecting drug use co-morbidity (e.g. from eastern Europe).
Minor comment: please spell out intimate partner violence (IPV) in the abstrct
